# Three Logistic Predictive Models for the Prediction of Mortality and Major Pulmonary Complications after Cardiac Surgery

**DOI:** 10.3390/medicina59081368

**Published:** 2023-07-26

**Authors:** Elena Bignami, Marcello Guarnieri, Ilaria Giambuzzi, Cinzia Trumello, Francesco Saglietti, Stefano Gianni, Igor Belluschi, Nora Di Tomasso, Daniele Corti, Ottavio Alfieri, Marco Gemma

**Affiliations:** 1Anesthesiology, Critical Care and Pain Medicine Division, Department of Medicine and Surgery, University of Parma, Viale Gramsci 14, 43126 Parma, Italy; elenagiovanna.bignami@unipr.it; 2Department of Anesthesia and Intensive Care, Grande Ospedale Metropolitano Niguarda, 20162 Milan, Italy; stefano.gianni@ospedaleniguarda.it; 3Department of Cardiovascular Surgery, Centro Cardiologico Monzino-IRCCS, 20122 Milan, Italy; i.giambuzzi@hotmail.it; 4Department of Clinical and Community Sciences, DISCCO University of Milan, 20126 Milan, Italy; 5Department of Cardiac Surgery, IRCCS San Raffaele Scientific Institute, 20132 Milan, Italy; trumello.cinzia@hsr.it (C.T.); belluschi.igor@hsr.it (I.B.); alfieri.ottavio@hsr.it (O.A.); 6Department of Anesthesia and Intensive Care, Azienda Ospedaliera Santa Croce e Carle, 12100 Cuneo, Italy; saglietti.md@gmail.com; 7Department of Anesthesia and Intensive Care, IRCCS San Raffaele Scientific Institute, 20132 Milan, Italy; ditomasso.nora@hsr.it (N.D.T.); danielecorti85@gmail.com (D.C.); 8Intensive Care Unit, Department of Neurosurgery, Fondazione IRCCS Istituto Neurologico Carlo Besta, 20133 Milan, Italy

**Keywords:** cardiac anesthesia, mortality, postoperative pulmonary complications

## Abstract

*Background and Objectives:* Pulmonary complications are a leading cause of morbidity after cardiac surgery. The aim of this study was to develop models to predict postoperative lung dysfunction and mortality. *Materials and Methods:* This was a single-center, observational, retrospective study. We retrospectively analyzed the data of 11,285 adult patients who underwent all types of cardiac surgery from 2003 to 2015. We developed logistic predictive models for in-hospital mortality, postoperative pulmonary complications occurring in the intensive care unit, and postoperative non-invasive mechanical ventilation when clinically indicated. *Results:* In the “preoperative model” predictors for mortality were advanced age (*p* < 0.001), New York Heart Association (NYHA) class (*p* < 0.001) and emergent surgery (*p* = 0.036); predictors for non-invasive mechanical ventilation were advanced age (*p* < 0.001), low ejection fraction (*p* = 0.023), higher body mass index (*p* < 0.001) and preoperative renal failure (*p* = 0.043); predictors for postoperative pulmonary complications were preoperative chronic obstructive pulmonary disease (*p* = 0.007), preoperative kidney injury (*p* < 0.001) and NYHA class (*p* = 0.033). In the “surgery model” predictors for mortality were intraoperative inotropes (*p* = 0.003) and intraoperative intra-aortic balloon pump (*p* < 0.001), which also predicted the incidence of postoperative pulmonary complications. There were no specific variables in the surgery model predicting the use of non-invasive mechanical ventilation. In the “intensive care unit model”, predictors for mortality were postoperative kidney injury (*p* < 0.001), tracheostomy (*p* < 0.001), inotropes (*p* = 0.029) and PaO_2_/FiO_2_ ratio at discharge (*p* = 0.028); predictors for non-invasive mechanical ventilation were kidney injury (*p* < 0.001), inotropes (*p* < 0.001), blood transfusions (*p* < 0.001) and PaO_2_/FiO_2_ ratio at the discharge (*p* < 0.001). *Conclusions:* In this retrospective study, we identified the preoperative, intraoperative and postoperative characteristics associated with mortality and complications following cardiac surgery.

## 1. Background

Although some scores are available to quantify the risk of mortality and morbidity after cardiac surgery [1,2,3], it is still a problematic issue in the context of clinical decision making. In particular, postoperative pulmonary complications (PPC; complete definition in Appendix A) are still a leading cause of morbidity after cardiac surgery, requiring longer hospital and intensive care unit (ICU) stay [4], higher mortality and increased costs [5,6,7,8,9,10].

Various factors, including inflammatory response following cardiopulmonary bypass, transfusions, the suspension of mechanical ventilation and blood flow through pulmonary circulation during cardiopulmonary bypass, myocardial damage and hyperoxia can all contribute to post-cardiac surgery lung injury [11,12,13,14,15]. Moreover, many risk factors for lung dysfunction and prolonged mechanical ventilation after cardiac surgery have been identified. These include preoperative variables (age, sex, cardiovascular risk factors, chronic lung disease, chronic kidney disease and coexisting endocarditis), intraoperative variables (type of surgery, pump time, intervention time, transfusions and bleeding) and postoperative variables (inotrope dependency and cardiogenic shock) [16,17,18,19,20,21,22,23]. Pulmonary damage has also been associated with the need for non-invasive mechanical ventilation (NIMV), re-intubation, and readmission to the ICU [24,25].

Regarding the complexity of the variables involved in the development of pulmonary complications, help could be obtained from machine learning. Machine learning plays a crucial role in medical research, revolutionizing the way healthcare professionals analyze vast amounts of complex data and make informed decisions. By utilizing powerful algorithms and statistical models, machine learning algorithms can uncover patterns, trends, and correlations in medical data that would be challenging for humans to detect. This technology enables researchers to develop predictive models for disease diagnosis, prognosis, and treatment outcomes, thereby empowering personalized medicine [26]. Additionally, they aid in medical image analysis and in diagnostic algorithms, enhancing the accuracy and speed of diagnoses [27,28]. The continuous integration of machine learning in medical research has the potential to advance our understanding of diseases, improve patient care, and ultimately transform the field of healthcare.

The PaO_2_/FiO_2_ ratio is a widely used and helpful tool, particularly in non-cardiac surgery, for the assessment of lung injury according to the Berlin criteria [23,29,30]. However, few studies have investigated the predictive power of this parameter in cardiac surgery [7]. Moreover, predictive models with strong clinical implications for perioperative care are still lacking [10], especially for cardiac surgery and in the ICU.

The aim of this study was to investigate the predictive factors of three distinct endpoints. Our primary endpoint was in-hospital mortality, with our secondary endpoints being the need for NIMV after ICU discharge and the occurrence of PPC after cardiac surgery.

## 2. Materials and Methods

This was a single-center, observational, retrospective study that took place at a university hospital in Italy from 2003 to 2015, following the ethical guidelines of the 1975 Declaration of Helsinki [31]. We retrospectively analyzed clinical and administrative data. Following approval from the Ethics Committee of the San Raffaele IRCCS Scientific Institute (114/INT/2015, approval date 29 October 2015), 11,285 consecutive adult patients, with American Society of Anesthesiologists (ASA) physical status class II or higher, who had undergone open-heart surgery with or without cardiopulmonary by-pass and with or without aortic cross-clamping, were included in this study. Specifically, we extracted information from a large database that included all patients who were admitted to the cardiac surgery department. We included all available information concerning potential pulmonary complications and mortality after cardiac surgery, including preoperative, intraoperative, and postoperative data.

At our center, patients undergoing anesthesia of any kind routinely give informed written consent, which includes eventual inclusion in retrospective research studies. Given the retrospective nature of the study, the Ethics Committee accepted this standard consent as being sufficient and did not request a specific informed-consent form. Every patient included in our study gave informed consent to use these clinical data for research purposes. Every type of surgical intervention was included in this study. All patients underwent a pre-defined protocol for perioperative management. They were all premedicated with morphine and managed with balanced general anesthesia, either volatile anesthetics or total intravenous anesthesia, as preferred by the clinician. Intraoperative ventilation was set as follows: tidal volume of 8 mL/kg of ideal body weight with positive end-expiratory pressure at a minimum level of 5 cmH_2_O applied to all patients unless contraindicated (dynamic overinflation, hemodynamic instability, etc.). The respiratory rate was titrated to maintain normocapnia. A single alveolar recruitment maneuver was applied after cardiopulmonary bypass. Following surgery, patients were kept unconscious until they were hemodynamically stable and weaned with pressure support ventilation. Details on the routine perioperative management carried out at the center are summarized in Appendix A [17,32,33]. Since potentially predictive variables are observable at different moments during a patient’s clinical course, we built a first model for each endpoint considering only those variables available upon hospital admission (“preoperative model”), and a second model regarding variables available at the time of surgery (“surgery model”). A third model was created after surgery, including variables available during ICU stay (“ICU model”). This approach aimed to provide clinically applicable prognostic tools throughout the various stages of each patient’s clinical course.

The primary outcome of this study was in-hospital mortality. We studied the need for NIMV after ICU discharge and the occurrence of PPC after cardiac surgery as secondary outcomes.

### 2.1. Statistical Analysis

Statistical analysis was carried out using the dedicated software Stata 11.1 (Copyright 2009 Stata Corp. LP, Stata Corp, 4905 Lakeway Drive, College Station, Texas 77845, USA). Categorical variables are reported as numbers (percentages). Continuous variables are reported as mean ± SD. Their normal distribution was confirmed by both the Shapiro–Wilk W test for normality and the Kolmogorov–Smirnov tests of the equality of distribution.

We developed three logistic predictive models for in-hospital mortality (as a dichotomous variable): a first model (“preoperative model”) where the variable selection was applied only to variables available at the time of hospital admission, a second model (“surgery model”) that also considered variables that become available at the time of surgery, and a third model (“ICU model”) considering variables available upon ICU admission. We similarly developed three logistic predictive models indicating the need for postoperative NIMV (as a dichotomous variable). Moreover, we developed two logistic predictive models (“preoperative model” and “surgery model”) for the occurrence of any PPC in the ICU (as a dichotomous variable). Variable selection was performed with a stepwise method with a probability to enter 0.10 and a probability to exit 0.20.

Model accuracy was compared by generating non-parametric ROC curves for each model and comparing their AUROC with the algorithm proposed by DeLong ER et al. [34]. We reported AUROCs as AUROC (95% CI). Model calibration was assessed by reporting the intercept (“calibration-in-the-large”) and slope of the relevant calibration plots [34]. Marginal predictions were used for the calculation of overall validation measures.

Since the PaO_2_/FiO_2_ ratio entered our final predictive models both for in-hospital mortality and NIMV use given its widespread use as a predictor thereof, we studied the relevant ROC curves to report the PaO_2_/FiO_2_ cutoff value that maximizes sensitivity and specificity.

Our series covers a wide time span, so the year of surgery was considered a random effect variable and the previously mentioned analysis models that were were studied as mixed logistic regression models. The difference between mixed and logistic regression models was tested using the log-likelihood ratio test. The intra-cluster correlation coefficient (ICC) was reported for each mixed-effect model. Random intercepts were omitted from the prediction equation to provide risk estimates for “an average” year, which would be clinically useful when addressing prediction for years, but were not in our sample.

Moreover, given the large number of cases in our series, we split the series into a training set and a test set. The previously mentioned models were obtained from the training set and were thereafter tested on the test set. The training set was obtained by selecting cases from our original series with a Bernoulli process (random selection without replacement) with 0.66 probability; this generated a training set containing 7448 cases. The test set consisted of the 3837 cases not selected in the training set. The AUROC of the models applied to the test sets was reported together with their 95% CI and compared with the corresponding training test AUROCs using the Chi-square test.

In the variable selection process, variables with *p* < 0.20 were considered possible predictors for the multivariate model. When developing the final model, *p* < 0.05 was considered significant. The interaction was checked, and linearity in the logit was verified using the fractional polynomial method. Figure 1 shows the time points involved in the development of the predictive models.

In reporting the final models, odds ratios are shown as OR (95% CI). We calculated the best probability cutoff value for each model built with the training set (the cutoff value maximizing sensitivity and specificity) and provided the relevant sensitivity, specificity, positive and negative predicted values, and percentage of correctly classified cases. Under the definition of NIMV, we included all the techniques that involved the application of positive pressure using a face mask or helmet, mostly including continuous positive airway pressure (CPAP) and pressure-supported ventilation with positive end-expiratory pressure. The main indication for CPAP was a hypoxemic respiratory failure of any origin. The main indication for pressure-supported NIMV was respiratory acidosis.

### 2.2. Sample Size Calculation

Given the retrospective nature of our study, no formal prior sample size evaluation was performed. Nevertheless, in building our logistic regression models, we respected the rule derived from simulation studies [35] stating that a minimum of 10 events per parameter are needed to avoid problems of overestimated or underestimated variances. Although this rule might be too stringent a requirement [36], as far as we know there is no compelling evidence to discard it.

## 3. Results

Our study population included 11,285 cases, which are presented in Table 1 and Table 2, both globally, and divided between the training and test sets. During the same study period, a total of 14,267 patients underwent cardiac surgery at our center. However, as we were unable to obtain complete information on 2982 patients, so we decided to exclude them from the study.

The most frequent procedure (5564 cases (49.31%)) carried out was combined surgery (coronary artery bypass graft + valvular procedure), followed by valvular surgery alone (5178 cases (45.88%)). Coronary surgery alone was performed on 1814 patients (16.07%). Aortic surgery was performed on 420 patients (3.71%). The remaining 999 patients (8.85%) underwent other cardiac procedures, such as percutaneous surgery (Mitraclip^®^ implantation), tumor exeresis, Maze procedure for atrial fibrillation or patent foramen ovale closure. The PaO_2_/FiO_2_ ratio at ICU discharge was 292.6 ± 114.0, showing normal distribution. Initially, the population’s mean age was 67.55 years ± 13.97. The mean body mass index (BMI) was 25.46 kg/m^2^ ± 3.95. The mean preoperative ejection fraction was 56.41% ± 9.77.

Intraoperative inotropic support was necessary for 5339 patients (47.31%). Non-invasive mechanical ventilation, considering both CPAP and pressure support ventilation, was prescribed for 658 patients (5.83%). Globally, a PPC was either diagnosed or treated with NIMV in 821 patients (7.3%). The overall mortality was 2.1% (236 cases).

### 3.1. In-Hospital Mortality Predictive Models

#### 3.1.1. Preoperative Model

Four variables predicted in-hospital mortality in the preoperative multivariate logistic model (whole model *p* < 0.0001; Pseudo R^2^ = 0.10): age, ejection fraction (EF), NYHA class, and elective vs. emergency procedure. When the year of the procedure was considered a random effect, the resulting mixed effect model did not differ from the corresponding logistic regression model (likelihood ratio test *p* = 1.000; ICC < 0.01). Table 3 shows a version of the models where we set the year of surgery as a random intercept.

This model’s AUROC was 0.81 (95% CI = 0.76–0.85).

The probability cutoff value maximizing sensitivity and specificity was 0.014, yielding 70.37% sensitivity, 72.44% specificity, 3.58% PPV (precision), 99.41% NPV and 72.41% correctly classified cases.

When the model was applied to the test set, its AUROC was 0.79 (95% CI = 0.75–0.83), which did not differ from the training test (*p* = 0.6685). The relevant calibration plot exhibited calibration-in-the-large = 0.0024 and calibration slope = 0.7065.

#### 3.1.2. Surgery Model

Five variables predicted in-hospital mortality in the surgery multivariate logistic model (whole model *p* < 0.0001; Pseudo R^2^ = 0.18): the use of inotropic drugs, the use of an intra-aortic balloon pump (IABP), age, NYHA class, and elective vs. emergency procedure. When the year of the procedure was considered a random effect, the resulting mixed effect model did not differ from the corresponding logistic regression model (likelihood ratio test *p* = 0.32; ICC = 0.02). Nevertheless, Table 3 shows the model where the year of surgery was a random intercept.

This model’s AUROC was 0.86 (95% CI = 0.82–0.89).

The probability cutoff value maximizing sensitivity and specificity was 0.016, yielding 78.00% sensitivity, 75.26% specificity, 4.92% PPV (precision), 99.52% NPV and 75.31% correctly classified cases.

When the model was applied to the test set, its AUROC was 0.85 (95% CI = 0.82–0.89): this did not differ from the training test (*p* = 0.7417), which was significantly higher than the corresponding test preoperative model’s AUROC (*p* = 0.0005). The relevant calibration plot exhibited calibration-in-the-large = 0.0057 and calibration slope = 0.8643 (Figure 2).

#### 3.1.3. ICU Model

Six variables predicted in-hospital mortality in the ICU multivariate logistic model (whole model *p* < 0.0001; Pseudo R^2^ = 0.26): peak serum creatinine value in the ICU, tracheostomy, use of inotropic drugs, NYHA class, age, and PaO_2_/FiO_2_ at ICU discharge. When the year of the procedure was considered a random effect, the resulting mixed-effect model did not differ from the corresponding logistic regression model (likelihood ratio test *p* = 1.000; ICC = 0.11). Nevertheless, Table 3 shows the model where the year of surgery was a random intercept.

This model’s AUROC was 0.90 (95% CI = 0.87–0.94).

The probability cutoff value maximizing sensitivity and specificity was 0.010, yielding 80.77% sensitivity, 80.59% specificity, 3.54% PPV (precision), 99.79% NPV and 80.59% correctly classified cases.

When the model was applied to the test set, its AUROC was 0.89 (95% CI = 0.84–0.93), which did not differ from the training test’s (*p* = 0.4390), but was significantly higher than the corresponding test surgery model’s AUROC (*p* = 0.0061). The relevant calibration plot exhibited calibration-in-the-large = 0.0021 and calibration slope = 0.8443 (Figure 2).

### 3.2. Postoperative NIMV Predictive Models

#### 3.2.1. Preoperative/Surgery Model

No variable with data available after surgery entered a multivariate model, hence the “Preoperative” and “Surgery” multivariate logistic models for NIMV use did not differ from one another.

Four variables predicted NIMV use in the preoperative/surgery multivariate logistic model (whole model *p* < 0.0001; Pseudo R^2^ = 0.05): age, EF, BMI and preoperative serum creatinine. When the year of the procedure was considered a random effect, the resulting mixed effect model differed from the corresponding logistic regression model (likelihood ratio test *p* = 0.0002; ICC = 0.08). Table 3 shows the model where the year of surgery was a random intercept.

This model’s AUROC was 0.71 (95% CI = 0.67–0.75).

The probability cutoff value maximizing sensitivity and specificity was 0.025, yielding 66.02% sensitivity, 64.40% specificity, 4.32% PPV (precision), 98.73% NPV and 64.44% correctly classified cases.

When the model was applied to the test set, its AUROC was 0.71 (95% CI = 0.67–0.75), which did not differ from the training test (*p* = 0.8687). The relevant calibration plot exhibited calibration-in-the-large = 0.0057 and calibration slope = 0.8643.

#### 3.2.2. ICU Model

Four variables predicted NIMV use after the discharge from the ICU multivariate logistic model (whole model *p* < 0.0001; Pseudo R^2^ = 0.13): peak serum creatinine in the ICU, inotropic drug use, P_a_O_2_/F_i_O_2_ ratio at ICU discharge, the use of blood products and BMI. When the year of the procedure was considered a random effect, the resulting mixed effect model (both intercept and slope) differed from the corresponding logistic regression model (likelihood ratio test *p* < 0.001; ICC = 0.17). Table 3 shows the model, where the year of surgery was a random intercept.

This model’s AUROC was 0.81 (95% CI = 0.77–0.85).

The probability cutoff value maximizing sensitivity and specificity was 0.048, yielding 71.72% sensitivity, 71.87% specificity, 10.91% PPV (precision), 98.14% NPV and 71.86% correctly classified cases.

When the model was applied to the test set, its AUROC was 0.79 (95% CI = 0.77–0.81), which did not differ from the training test (*p* = 0.3966) but was significantly higher than the corresponding test surgery model’s AUROC (*p* < 0.0001). The relevant calibration plot exhibited calibration-in-the-large = 0.0063 and calibration slope = 0.8814 (Figure 2).

### 3.3. Postoperative Pulmonary Complication Predictive Model

#### 3.3.1. Preoperative Model

Four variables predicted PPC in the preoperative multivariate logistic model (whole model *p* < 0.00001; Pseudo R^2^ = 0.06): chronic obstructive pulmonary disease (COPD), preoperative serum creatinine, EF and NYHA class. When the year of the procedure was considered a random effect, the resulting mixed effect model did not differ from the corresponding logistic regression model (likelihood ratio test *p* = 0.1767; ICC = 0.06). Table 3 shows the model, where the year of surgery was a random intercept.

This model’s AUROC was 0.70 (95% CI = 0.62–0.78).

The probability cutoff value maximizing sensitivity and specificity was 0.013, yielding 62.16% sensitivity, 62.11% specificity, 2.45% PPV (precision), 99.07% NPV and 62.11% correctly classified cases.

When the model was applied to the test set, its AUROC was 0.69 (95% CI = 0.61–0.76), which did not differ from the training test (*p* = 0.9299). The relevant calibration plot exhibited calibration-in-the-large = 0.0059 and calibration slope = 0.5278.

#### 3.3.2. Surgery Model

Four variables predicted PPC in the surgery multivariate logistic model (whole model *p* < 0.0001; Pseudo R^2^ = 0.088): the use of inotropic drugs, the use of IABP, COPD and preoperative serum creatinine. When the year of the procedure was considered a random effect, the resulting mixed effect model (both intercept and slope) differed from the corresponding logistic regression model (likelihood ratio test *p* = 0.0275; ICC = 0.08). Table 3 shows the model, where the year of surgery was a random intercept.

This model’s AUROC was 0.70 (95% CI = 0.62–0.78).

The probability cutoff value maximizing sensitivity and specificity was 0.016, yielding 72.50% sensitivity, 65.25% specificity, 2.95% PPV (precision), 99.39% NPV and 65.35% correctly classified cases.

When the model was applied to the test set, its AUROC was 0.68 (95% CI = 0.60–0.76), which did not differ from the training test’s AUROC (*p* = 0.3478) but was not significantly different from the corresponding test preoperative model’s AUROC (*p* = 0.4467). The relevant calibration plot exhibited calibration-in-the-large = −0.0006 and calibration slope = 1.1434 (Figure 2).

### 3.4. ROC Curve Analysis of the PaO_2_/FiO_2_ Ratio

A ROC curve was developed using the PaO_2_/FiO_2_ ratio at ICU discharge and the incidence of NIMV use during hospital stay. The PaO_2_/FiO_2_ ratio value, maximizing sensitivity and specificity was 239 mmHg. At this point in the curve, sensitivity was 66.53% while specificity was 66.06%. Correct classification occurred in 66.51% of cases. Figure 3 shows the ROC curve we developed.

The outcome data of the whole population are summarized in Table 4.

## 4. Discussion

We investigated the potential predictive factors for mortality, NIMV use, and PPC occurrence after cardiac surgery in a large adult population based on pre-operative, intra-operative and post-operative variables (Table 3).

Our three models offer a comprehensive view of the journey of patients undergoing cardiac surgery from preoperative evaluation up to ICU discharge, with a stepwise use of new variables. This approach is different from the other models available in the literature. The robustness of the underlying statistics might offer clinicians a powerful tool for the prediction of complications in the context of cardiac surgery.

The risk factors we underlined need to be taken into consideration during clinical decision-making, and patients at high risk of postoperative pulmonary dysfunction might benefit from a specific protocol for intraoperative and postoperative optimization, ICU discharge, respiratory chest physiotherapy and NIMV in the ward. It might also help decide the optimal timing for ICU discharge. Delayed discharge can lead to healthcare-related infections and is globally associated with a worse outcome [22] and increased costs. Premature discharge, however, might result in the need for readmission to the ICU, which can also be very harmful [25].

The literature describes numerous models for calculating mortality and morbidity in patients undergoing cardiac surgery, the most important being the EUROSCORE II and STS scores [37,38]. However, there are some limitations regarding their use, such as the small number of surgical cases included in the STS score [1] or the deteriorated prediction of higher-risk tertiles in EUROSCORE II [2].It is important, therefore, to continue searching for new models to guarantee a more patient-centered tailored approach. Indeed, our model is the first to be dynamic; new variables can be added that modify the prediction of risk regarding mortality and morbidity as patients progress from the preoperative phase to the operative phase, and finally to the postoperative phase. Of the other scores concerning PPC with extensive validation, the ARISCAT score [7] and the LAS VEGAS score [3] require mentioning. The peculiarity of these scores is that they are general and thus applicable to every kind of anesthesia. On the other hand, they do not focus on specificities for cardiac surgery, which we aimed to include in our study.

Another interesting point of the secondary analysis concerns the PaO_2_/FiO_2_ ratio at ICU discharge. The ROC curve we developed found a cutoff point maximizing the sensitivity and specificity of the PaO_2_/FiO_2_ ratio at 239. This information might help clinicians distinguish between those patients with sub-optimal gas exchange that require respiratory support and those who do not.

Other studies have investigated post-cardiac surgery pulmonary complications and the predictive value of the PaO_2_/FiO_2_ ratio. A prospective observational trial on 2725 consecutive cardiac surgery patients [24] identified lower PaO_2_/FiO_2_ ratio values in non-survivors compared with survivors and highlighted the importance of this parameter in predicting a worse postoperative outcome. Another smaller observational trial identified several risk factors for a reduced PaO_2_/FiO_2_ ratio during the postoperative period, including age, obesity, reduced cardiac function, emergency surgery, high creatinine levels, and inotropic support [7].

We also identified factors associated with the need for NIMV and the incidence of PPC. We confirmed the data provided by a retrospective trial [19] regarding obesity and postoperative hypoxia, defining preoperative BMI as a predictor for postoperative NIMV and a reduced PaO_2_/FiO_2_ ratio at ICU discharge. A smaller observational trial [18] identified BMI as an important risk factor, with findings similar to ours concerning the intraoperative risk factors for difficult respiratory management, concentrating on prolonged mechanical ventilation in the ICU. Finally, a recent study on 145 adult cardiac surgery patients showed a correlation between lower PaO_2_/FiO_2_ ratios and length of hospital and ICU stay [39].

The importance of our data is represented by a larger sample size compared with previous studies, the robustness of our statistical analysis, and the step-by-step inclusion of preoperative, intraoperative, and postoperative factors in a manner closely related to a clinical practice model, which might prove useful to identify high-risk patients that could benefit from closer monitoring and observation. Moreover, the need for NIMV and the presence of PPC following cardiac surgery have never been examined using a comprehensive predictive model. Finally, we were able to provide a new cutoff for the PaO_2_/FiO_2_ ratio evaluation in routine clinical activity.

The main limitation of the present study is that it was a retrospective analysis. We were unable to collect all the necessary data from each of the included patients, as many of the source documents were 12 years old, and it was at times difficult to obtain the correct information. Due to the stepwise selection, the global sample size was numerically inferior to the initial sample size. We acknowledge that the overall incidence of PPCs, even if grouped with the application of NIMV, is quite low compared with the incidence reported in the literature [40].

We acknowledge that some random error may still be present in data coming from different years and multiple sources, but we are confident that the high number of patients analyzed and the robustness of the models we developed can overcome this limitation. Moreover, in the context of a large sample size, we recognize that the exclusion of 2982 patients might have introduced a selection bias that we were unable to quantify or eliminate.

One methodological limitation is the definition of the PPCs used, which is not the most recent provided by Abbott et al. [41]. Moreover, we do not have data regarding the number of patients treated with NIMV or CPAP before surgery, which might have slightly altered our main results.

Another important limitation is that anesthesia and surgical techniques have changed over the years, and we are aware that what we did in 2003 may not be the same as what we are doing nowadays. Risk factors such as preoperative conditions, intraoperative inotropes, mechanical support, postoperative infections, etc. are ever-present problems. The management and understanding of high-risk patients and complications might have changed, but the disease pathophysiology with its risk factors and predictors most probably has not.

## 5. Conclusions

Our results demonstrated that a lower PaO_2_/FiO_2_ ratio was linked to an increased risk of in-hospital mortality following cardiac surgery. This indicates that compromised oxygenation levels, as indicated by a lower ratio, can be a significant factor in determining the prognosis and survival of patients undergoing cardiac procedures.

Overall, our study highlights the importance of the PaO_2_/FiO_2_ ratio as a valuable predictive tool for assessing the risk of pulmonary complications, the need for non-invasive mechanical ventilation, and hospital mortality in patients undergoing cardiac surgery. These findings provide healthcare professionals with valuable insights into risk stratification and individualized patient management, ultimately leading to improved clinical decision-making and better outcomes for cardiac surgery patients.

## Figures and Tables

**Figure 1 medicina-59-01368-f001:**
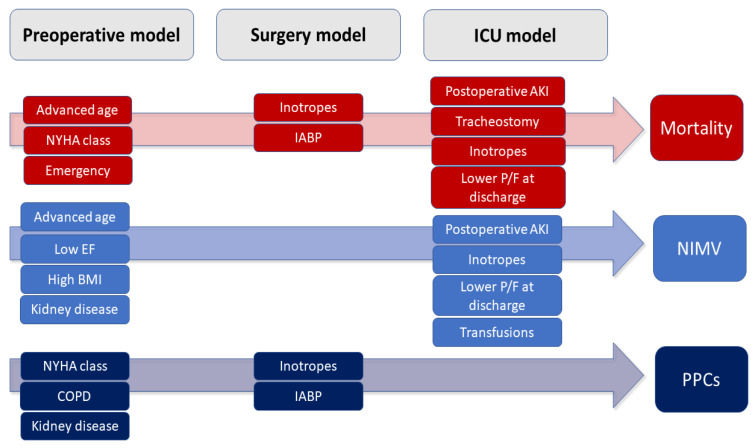
The timelines summarize the pathway of patients undergoing cardiac surgery, with a focus on the variables showing the statistical significance of the major endpoints of this study. List of abbreviations. NYHA: New York Heart Association; EF: ejection fraction; BMI: body mass index; COPD: chronic obstructive pulmonary disease; IABP: intra-aortic balloon pump; AKI: acute kidney injury; P/F: PaO_2_/FiO_2_ ratio; NIMV: non-invasive mechanical ventilation; PPCs: postoperative pulmonary complications.

**Figure 2 medicina-59-01368-f002:**
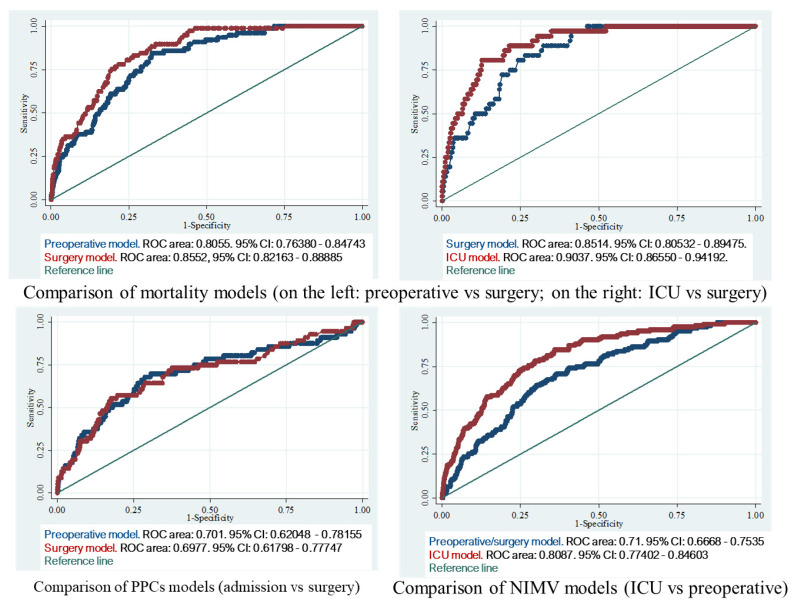
Comparison between predictive models and ROC curves. The first figure (**top left**) shows the comparison between the preoperative model (blue line) and the surgery model (red line) for mortality, outlining an increase in the ROC area from the first to the second model. The second figure (**top right**) shows a further increase in the ROC area between the surgery model for mortality (blue line) and the ICU model (red line). The third figure (**bottom left**) shows the ROC curve comparison between the preoperative (blue line) and surgery (red line) model for the development of postoperative pulmonary complications. Finally, the fourth figure (**bottom right**) shows the ROC curve comparison between the preoperative/surgery model (blue line) for the application of NIMV in the ward and the ICU model (red line). List of abbreviations. ICU: intensive care unit; PPCs: postoperative pulmonary complications; NIMV: non-invasive mechanical ventilation.

**Figure 3 medicina-59-01368-f003:**
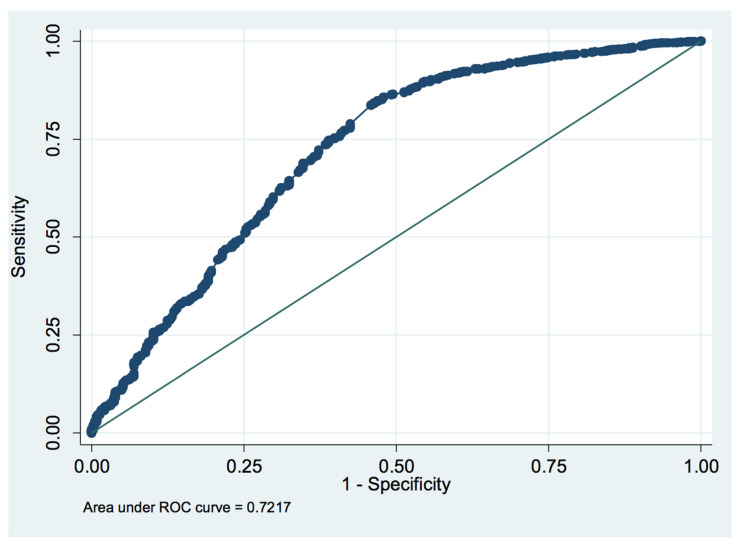
The ROC curve for the PaO_2_/FiO_2_ ratio and the incidence of post-operative non-invasive mechanical ventilation. The PaO_2_/FiO_2_ ratio value that maximizes sensitivity and specificity is 239 mmHg. The sensitivity was 66.53%, while the specificity was 66.06%.

**Table 1 medicina-59-01368-t001:** Description of the baseline data of the study population (N = 11,285). Data are expressed as mean ± standard deviation or number (percentage). Preoperative chronic obstructive pulmonary disease (COPD) was defined according to clinical criteria (patients were considered to be affected by COPD if they were prescribed inhalation bronchodilators, corticosteroids or other drugs labeled for COPD therapy by a pneumologist or the family physician, even without obtaining confirmation from preoperative instrumental data) at the time of hospital admission.

Variable	Training Set (7448 (66%) Cases)	Test Set (3837 (34%) Cases)
Age, yrs	67.98 ± 14.00	67.59 ± 13.98
Male sex, n (%)	4394 (59%)	2302 (60%)
BMI, kg/m^2^	25.45 ± 3.97	25.46 ± 3.00
Preoperative EF, %	56.23 ± 9.00	56.19 ± 9.99
NYHA class > II, n (%)	(28.20%)	1082 (28.20%)
Preoperative comorbidities		
COPD, n (%)	476 (6.39%)	252 (6.56%)
Hypertension, n (%)	3706 (49.75%)	2003 (52.19%)
Type II Diabetes, n (%)	779 (10.46%)	426 (11.10%)
Preoperative creatinine, mg/dL	0.98 ± 0.68	0.98 ± 0.66
Chronic renal failure, n (%)	1418 (19%)	789 (20%)
Peripheral vasculopathy, n (%)	1089 (14.63%)	518 (13.50%)
Smoking habits, n (%)	1,279,504 (17.17%)	653 (17.01%)
Stroke, n (%)	(6.76%)	279 (7.27%)
Timing of surgery		
Emergency or urgency, n (%)	63 (0.85%)	32 (0.85%)
Planned, n (%)	7385 (99.15%)	3805 (99.15%)
Type of surgery		
Valvular surgery, n (%)	2944 (39.53%)	1535 (39.99%)
Coronary surgery, n (%)	991 (13.30%)	476 (12.42%)
Ascending aorta aneurysm surgery, n (%)	236 (3.17%)	106 (2.77%)
Other surgical procedures, n (%)	343 (4.60%)	172 (4.48%)
Combined surgery (two or more procedures), n (%)	2934 (39.40%)	1548 (40.35%)

BMI—body mass index; EF—ejection fraction; COPD—chronic obstructive pulmonary disease.

**Table 2 medicina-59-01368-t002:** Description of the baseline data of the study population (N = 11,285). Data are expressed as mean ± standard deviation or number (percentage). Preoperative chronic obstructive pulmonary disease (COPD) was defined according to clinical criteria (patients were considered to be affected by COPD if they were prescribed inhalation bronchodilators, corticosteroids or other drugs labeled for COPD therapy by a pneumologist or the family physician, even without obtaining confirmation from preoperative instrumental data) at the time of hospital admission.

Variable	Preoperative Value
Age, y	67.55 ± 13.97
Male sex, n (%)	9844 (87.2%)
Height, cm	169 ± 9
Weight, kg	73 ± 13
BMI, kg/m^2^	25.46 ± 3.95
Preoperative EF, %	56.41% ± 9.77
NYHA class > II, n (%)	
Preoperative comorbidities	
COPD, n (%)	920 (8.1%)
Hypertension, n (%)	7.248 (64.2%)
Type II Diabetes, n (%)	1.528 (13.5%)
Preoperative creatinine, mg/dL	0.98 ± 0.67
Chronic renal failure, n (%)	1161 (10.2%)
Peripheral vasculopathy, n (%)	2036 (18.0%)
Smoking habits, n (%)	2443 (21.6%)
Stroke, n (%)	989 (8.8%)
Timing of surgery	
Emergency or urgency, n (%)	214 (1.9%)
Planned, n (%)	11,071 (98.1%)
Surgery type	
Valvular surgery, n (%)	5178 (45.88%)
Coronary surgery, n (%)	1814 (16.07%)
Ascending aorta aneurysm surgery, n (%)	420 (3.71%)
Other surgical procedures, n (%)	999 (8.85%)
Combined surgery (two or more procedures), n (%)	5564 (49.31%)

BMI—body mass index; EF—ejection fraction; COPD—chronic obstructive pulmonary disease.

**Table 3 medicina-59-01368-t003:** Results of the logistic regression models.

	Models for Mortality				Models for NIMV				Models for PPC			
Preoperative models	**Predictive Variable**	**Odds Ratio**	**95% CI**	***p*-Value**	**Predictive Variable**	**Odds Ratio**	**95% CI**	***p*-Value**	**Predictive Variable**	**Odds Ratio**	**95% CI**	***p*-Value**
	Age	1.05	1.02–1.08	<0.001	Age	1.04	1.02–1.06	<0.001	COPD	2.63	1.31–5.28	0.007
	Preoperative EF	0.97	0.95–0.99	0.011	Preoperative EF	0.97	0.96–1.00	0.023	Creatinine	1.48	1.19–1.83	<0.001
	NYHA class	2.97	1.63–5.41	<0.001	BMI	1.10	1.05–1.15	<0.001	EF	0.97	0.95–0.99	0.004
	Elective surgery	0.29	0.90–0.91	0.036	Preoperative Creatinine	1.26	1.01–1.58	0.043	NYHA class	1.81	1.05–3.14	0.033
	Random effect variable	SD	SE	*p*	Random effect variable	SD	SE	*p*	Random effect variable	SD	SE	*p*
	Year of surgery	<0.001	0.37	1.000	Year of surgery	0.53	0.18	<0.001	Year of surgery	0.28	0.20	0.176
Surgery models	**Predictive Variable**	**Odds Ratio**	**95% CI**	***p*-Value**	**Predictive Variable**	**Odds Ratio**	**95% CI**	***p*-Value**	**Predictive Variable**	**Odds Ratio**	**95% CI**	***p*-Value**
	Inotropes in the operating room	3.09	1.45–6.6	0.003	Age	1.04	1.02–1.06	<0.001	Inotropes in the operating room	2.79	1.38–5.64	0.004
	IABP in the operating room	3.91	1.90–8.04	<0.001	Preoperative EF	0.97	0.96–1.00	0.023	IABP in the operating room	2.64	1.02–6.81	0.045
	Age	1.06	1.03–1.10	<0.001	BMI	1.10	1.05–1.15	<0.001	COPD	3.74	1.64–8.51	0.002
	NYHA class	2.35	1.24–4.47	0.009	Preoperative Creatinine	1.26	1.01–1.58	0.043	Preoperative creatinine	1.39	1.07–1.81	0.014
	Elective surgery	0.22	0.08–0.65	0.006								
	Random effect variable	SD	SE	*p*	Random effect variable	SD	SE	*p*	Random effect variable	SD	SE	*p*
	Year of surgery	0.24	0.30	0.320	Year of surgery	0.53	0.18	<0.001	Year of surgery	0.53	0.25	1.329
ICU models	**Predictive Variable**	**Odds Ratio**	**95% CI**	***p*-Value**	**Predictive Variable**	**Odds Ratio**	**95% CI**	***p*-Value**				
	Creatinine peak	1.50	1.24–1.82	<0.001	Creatinine peak	1.35	1.21–1.51	<0.001				
	Tracheostomy	18.08	7.14–45.76	<0.001	Inotropes	1.60	1.25–2.04	<0.001				
	Inotropes in the ICU in the ICU	2.52	1.01–5.77	0.029	P/F	0.99	0.991–0.993	<0.001				
	NYHA class	2.79	1.35–5.78	0.006	Blood transfusion	2.41	1.87–3.13	<0.001				
	Age	1.08	1.03–1.12	<0.001	BMI	1.07	1.05–1.11	<0.001				
	P/F ratio	0.1	0.99–0.1	0.028								
	Random effect variable	SD	SE	*p*	Random effect variable	SD	SE	*p*				
	Year of surgery	<0.001	0.44	1.000	Year of surgery	0.83	0.18	<0.001				

List of abbreviations used: CI: confidence interval; SD: standard deviation; SE: standard error; EF: ejection fraction; NYHA: New York Heart Association; BMI: body mass index; COPD: chronic obstructive pulmonary disease; IABP: intra-aortic balloon pump; ICU: intensive care unit; P/F: P_a_O_2_/F_i_O_2_ ratio at the discharge from the ICU.

**Table 4 medicina-59-01368-t004:** Study outcomes in the ICU for the study population. The values are expressed as n (%), median interquartile range where appropriate.

Outcome	Incidence
Overall mortality, n (%)	236 (2.1%)
Postoperative pulmonary complications, n (%)	213 (1.9%)
Postoperative pulmonary complications (including the use of NIMV), n (%)	821 (7.3%)
Need for NIMV before hospital discharge, n (%)	609 (5.4%)
Re-intubation, n (%)	97 (0.86%)
Inotropes, n (%)	3809 (33.75%)
Intra-aortic balloon pump, n (%)	395 (3.5%)
Blood products, n (%)	2834 (18%)
Renal replacement therapy, n (%)	175 (1.55%)
VA-ECMO support, n (%)	12 (0.1%)
Septic shock, (%)	56 (0.5%)
Length of ICU stay, days	1 (1–3)
Length of hospital stay, days	6 (5–8)

List of abbreviations: NIMV: Non-invasive mechanical ventilation, ECMO: extracorporeal membrane oxygenation, ICU: intensive care unit.

## Data Availability

The data presented in this study are available upon request from the corresponding author. The data are not publicly available due to privacy reasons. All data generated or analyzed during this study are included in this published article (and its Appendix A).

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
