# Peer review of "Three Logistic Predictive Models for the Prediction of Mortality and Major Pulmonary Complications after Cardiac Surgery"

_medicina, 2023, doi:10.3390/medicina59081368_

Round 1
Reviewer 1 Report
I had the possibility to review the manuscript titled "Three logistic predictive models for the prediction of mortality and major pulmonary complications after cardiac surgery" written by Bignami et al.
Overall, I would like to acknowledge the significance of the issues analyzed in this retrospective study. Patient prognostication and mortality prediction are topics of constant interest among active clinicians, and even though numerous studies have attempted to facilitate decision-making regarding patient management in the perioperative period, there is still a paucity of data for the actual predictive variables for patient outcomes.
I find the manuscript well-written and comprehensible, with a good scientific background and the limitations described.
The minor issue I have with the manuscript regards the repetition of the predictive models' description in the "Methodology" section of the study. The models are described in an almost identical manner in lines "114-123" and "148-159", with earlier but more condensed version mentioned also in lines "99-104". I would advise the authors to keep one detailed description of the three predictive models in the "Statistical analysis" section of the manuscript and avoid its unnecessary duplication later in the same section, above Figure 1.
Reviewer 2 Report
Dear authors, the article submitted is novel and interesting. Please make the changes and resubmit.
Abstract: Okay
Background: This section has to be improved. Please add your contributions in points along with research gaps.
Please mention about machine learning and its uses. Literature review is missing. Machine learning has been used to diagnose various diseases. Please include a few of them. The following related articles could be included:
1. Chadaga K, Prabhu S, Bhat V, Sampathila N, Umakanth S, Chadaga R. A Decision Support System for Diagnosis of COVID-19 from Non-COVID-19 Influenza-like Illness Using Explainable Artificial Intelligence. Bioengineering. 2023 Mar 31;10(4):439.
2. Khanna VV, Chadaga K, Sampathila N, Prabhu S, Bhandage V, Hegde GK. A distinctive explainable machine learning framework for detection of polycystic ovary syndrome. Applied System Innovation. 2023 Feb 23;6(2):32.
3. Chadaga K, Prabhu S, Sampathila N, Nireshwalya S, Katta SS, Tan RS, Acharya UR. Application of artificial intelligence techniques for monkeypox: a systematic review. Diagnostics. 2023 Feb 21;13(5):824.
Methods: Please include a table regarding descriptive statistical measures. Please create a table mentioning the attributes and its signficance.
Labelling of the sections is not made properly. Please check.
You need to compare between preoperative model, surgery model, ICU model
You need to make a table and include all the results. The table should consist of columns such as accuracy, precision, recall and other metrics. Please refer the papers mentioned above.
Conclusions: two lines conclusions are not enough. Please add a paragraph or two.
Overall verdict: The manuscript is good, however, improvements are needed for it to be published in a prestigious journal such as Medicina!
